# Tide-Surge-Wave Interaction in the Taiwan Strait during Typhoons Soudelor (2015) and Dujuan (2015)

**Li Zhang [1,2], Shaoping Shang [1,2,3,\*], Feng Zhang [1,2,\*] and Yanshuang Xie [1,2,3]**

[1]   College of Ocean & Earth Sciences, Xiamen University, Xiamen 361005, China; zhangnier99@sina.com (L.Z.); xys_0614@126.com (Y.X.)

[2]   Key Laboratory of Underwater Acoustic Communication and Marine Information Technology, Xiamen University, Ministry of Education, Xiamen 361005, China

[3]   Center for Ocean Observation Technologies, Xiamen University, Xiamen 361005, China

\*   Correspondence: spshang@xmu.edu.cn (S.S.); zhang168feng@gmail.com (F.Z.); Tel.: +86-131-2339-3626 (S.S.)

**Abstract:** Typhoons Soudelor (2015) and Dujuan (2015) were two of the strongest storms to affect the Taiwan Strait in 2015. This study investigated the response of the waters on the western bank of the Taiwan Strait to the passage of Soudelor and Dujuan. This included an investigation of the resonant coupling between the tide and storm surge, typhoon wave variation caused by the storm tide, and wave-induced water level rise. Analyses conducted using numerical model simulations and observations from tidal stations and buoys, obtained during the passage of both Soudelor and Dujuan, revealed that resonant coupling between the astronomical tide and storm surge in the Taiwan Strait was prominent, which resulted in tidal period oscillation on the storm surge and reduced tidal range. The tide wave arrived earlier than the predicted astronomical tide because of the existence of the storm surge, which was attributable to acceleration of the tidal wave caused by the water level rise. Wave height observations showed that the storm tide predominantly affected the waves, which resulted in wave heights that oscillated within the tidal period. Numerical experiments indicated that both the current and the water level affected wave height. Waves were affected mainly by the current in the middle of the Taiwan Strait, but mostly by water level when the water level was comparable with water depth. Wave setup simulations revealed that wave setup also oscillated within the tidal period, and that local bathymetry was the most important influencing factor of wave setup distribution.

**Keywords:** typhoon; coastal water responses; storm surge; waves; tide–surge interaction

## 1. Introduction

The Taiwan Strait is located on the wide continental shelf of the East China Sea. The long narrow strait with average depth of nearly 60 m lies between the Chinese mainland and the island of Taiwan. Water from the North Pacific is mixed with water from the Luzon Strait in the Taiwan Strait. Regionally, the semidiurnal tide is dominant. The tidal range of 2–4 m from south to north, attributable to blocking by the island of Taiwan and the channeling effect of the Taiwan Strait, results in extremely strong tidal currents [1].

The Taiwan Strait is in an area prone to strong typhoons. Typhoons in this area often induce storm surges, high waves, and coastal inundation, which represent some of the most serious natural hazards to affect the lives and property of the regional population. During 1949–2005, 60% of the typhoons that passed through the Taiwan Strait area made landfall twice, first on the island of Taiwan and then on the western bank of the Taiwan Strait [2].

The interaction between tide and storm surge has been studied since the 1980s. Both Wolf [2] and Horsburgh and Wilson [3] analyzed tidal observations and concluded that increased water depth attributable to storm surge induces acceleration of tidal wave propagation. Park and Suh [4] reported that storm surge is inversely proportional to water depth and tide, and that coupling of storm surge and tides is inevitable in coastal areas with shallow water depth and large tides. Based on study of the interaction between tides and storm surges using numerical simulations and observations, Bernier and Thompson [5] concluded that nonlinear bottom stress is an important factor. Zhang et al. [6] studied tide–surge interaction in the Taiwan Strait during the passage of Typhoon Dan (1999). They found that tide–surge interaction in the northern portion of the Taiwan Strait is intensified by the features of the strait, and that nonlinear bottom stress is an important factor that must be considered when predicting the oscillation due to tide–surge interaction.

The interaction between typhoon-induced waves and sea level elevation induced by tides and storm surges has been studied previously. According to Wolf [2], the effect of tides and surges on waves must be considered in coastal areas because wave height is controlled largely by water depth, and because waves contribute to the total water level by means of the wave setup through wave radiation stress. Based on investigation of wave–tide interaction in the Yellow and East China seas, Moon [7] concluded that tides were the most influential factor with regard to modulation of regional mean wave characteristics. According to the findings of Chen et al. [8] in relation to the east coast of Taiwan, Huang et al. [9] with regard to Tampa Bay (Florida, USA), and Bertin et al. [10] in relation to the Bay of Biscay, waves should not be neglected in storm surge modeling in areas where large wave heights occur and the sea bottom slope is steep.

The objective of this study was to investigate the response of the water of the Taiwan Strait (primarily on the western bank) to interactions among tides, storm surges, and typhoon-induced waves during the passage of typhoons Soudelor and Dujuan. A state-of-the-art unstructured grid storm surge–wave–tide coupled model was applied to simulate typhoon-induced storm surges for Soudelor and Dujuan. The storm surge–wave–tide coupled model is described in Sections 2 and 3 presents model validation for the tides, surges, and significant waves during the storms. Analysis of the interaction between water level and waves is provided in Section 4. Finally, our conclusions are presented in Section 5.

## 2. Materials and Methods

### 2.1. FETSWCM

The tide and storm surge were simulated using the FETSWCM (finite element tide–surge–wave coupled model) model [11,12]. The FETSWCM model is a two-dimensional hydrodynamic model encompassed with a specific typhoon wind model for the Taiwan Strait [13]. FETSWCM has demonstrated satisfactory performance in storm surge forecasting along the Fujian Coast, which is on the western bank of the Taiwan Strait. Using a right-handed Cartesian coordinate system, combined GWCE (generalized wave continuity equation) [14] and VIMEs (vertically integrated momentum equations), the governing equations of FETSWCM in two-dimensional form are given as:

$$\frac{\partial^2 \varsigma}{\partial t^2} + \tau_0 \cdot \frac{\partial \varsigma}{\partial t} + \frac{\partial}{\partial x}\left(\frac{\partial(U \cdot H)}{\partial t} + \tau_0 \cdot U \cdot H\right) + \frac{\partial}{\partial y}\left(\frac{\partial(V \cdot H)}{\partial t} + \tau_0 \cdot V \cdot H\right) - U \cdot H \cdot \frac{\partial \tau_0}{\partial x} - V \cdot H \cdot \frac{\partial \tau_0}{\partial y} = 0,$$

(1)

$$\frac{\partial U}{\partial t} + U \cdot \frac{\partial U}{\partial x} + V \cdot \frac{\partial U}{\partial y} - f \cdot V = -g \cdot \frac{\partial(\varsigma + P_s/g/\rho_0)}{\partial x} + \frac{\tau_{sx}}{H \cdot \rho_0} + \frac{\tau_{wx}}{H \cdot \rho_0} - \frac{\tau_{bx}}{H \cdot \rho_0},$$

(2)

$$\frac{\partial V}{\partial t} + U \cdot \frac{\partial V}{\partial x} + V \cdot \frac{\partial V}{\partial y} + f \cdot U = -g \cdot \frac{\partial(\varsigma + P_s/g/\rho_0)}{\partial y} + \frac{\tau_{sy}}{H \cdot \rho_0} + \frac{\tau_{wy}}{H \cdot \rho_0} - \frac{\tau_{by}}{H \cdot \rho_0},$$

(3)

where

$\zeta$ = the rise of the water level above the undisturbed sea level;

$t$ = time;

$x$, $y$ = longitude (east) and latitude (north), respectively;

$U$, $V$ = the east and north components, respectively, of depth averaged velocity;

$H$ = the total water depth, where $H = h + \zeta$;

$h$ = the depth of the undisturbed water;

$f$ = the Coriolis parameter;

$g$ = acceleration due to gravity;

$P_s$ = sea surface atmospheric pressure;

$\rho_a$ = the density of air;

$\rho_0$ = the density of sea water;

$W$ = the wind velocity vector at the height of 10 m above sea surface;

$\tau_{sx}$, $\tau_{sy}$ = the east and north components, respectively, of wind stress friction;

$$\tau_s = \rho_a \cdot C_s \cdot \left| \vec{W} \right| \cdot \vec{W} \tag{4}$$

where $C_s$ is the wind stress coefficient, which varies with wind velocity as follows:

$$C_s = \begin{cases} 1.052 \times 10^{-3} & \left| \vec{W} \right| \leq 6\ m/s \\ \left( 0.638 + 0.069 \left| \vec{W} \right| \right) \times 10^{-3} & 6\ m/s \leq \left| \vec{W} \right| \leq 30\ m/s \\ 2.708 \times 10^{-3} & \left| \vec{W} \right| \geq 30\ m/s \end{cases} \tag{5}$$

$\tau_{bx}$, $\tau_{by}$ = the east and north components, respectively, of bottom friction.

$$\tau_{bx} = \rho_0 \cdot C_d \cdot U \cdot \sqrt{(U^2 + V^2)} \tag{6}$$

$$\tau_{by} = \rho_0 \cdot C_d \cdot V \cdot \sqrt{(U^2 + V^2)} \tag{7}$$

where $C_d$ is the bottom friction coefficient, which was taken as 0.0015 in this study.

$\tau_{wx}$, $\tau_{wy}$ = the east and north components, respectively of wave force (gradient of the wave radiation stress) which are calculated through SWAN model (Section 2.2) as follows:

$$\tau_{wx} = -\frac{\partial S_{xx}}{\partial x} - \frac{\partial S_{xy}}{\partial y} \tag{8}$$

$$\tau_{wx} = -\frac{\partial S_{yx}}{\partial x} - \frac{\partial S_{yy}}{\partial y} \tag{9}$$

where $S_{xx}$, $S_{xy}$, $S_{yx}$, $S_{yy}$ = the radiation stress tensor (m/s).

In GWCE, the spatially differentiated momentum equation in its conservative form is combined with the temporally differentiated continuity equation, and the continuity equation is multiplied by the numerical parameter $\tau_0$, which is a constant that controls the balance of the wave equation and the primitive continuity equation. Extensive analysis of the GWCE has demonstrated that the scheme is stable [15].

## 2.2. SWAN

The SWAN is a wave model developed at Delft University of Technology (The Netherlands), which computes random, short-crested, wind-generated waves in coastal regions and inland waters.

SWAN has been used widely in many studies throughout the world, and is has demonstrated satisfactory performance in wave modeling in the Taiwan Strait [16–18].

SWAN uses a spectrum action balance equation to describe the generation of waves and evolution process in the near shore area as follows [19]:

$$\frac{\partial N}{\partial t} + \frac{\partial}{\partial x}C_x N + \frac{\partial}{\partial y}C_y N + \frac{\partial}{\partial \sigma}C_\sigma N + \frac{\partial}{\partial \theta}C_\theta N = \frac{S}{\sigma} \tag{10}$$

where

$N$ = wave action density (Joule/m$^2$/Hz);
$t$ = time (s);
$\sigma$ = frequency (Hz);
$\theta$ = wave propagation direction;
$C_x$, $C_y$ = the propagation velocities of wave energy in spatial x-, y-space;

$$C_x = \frac{1}{2}\left[1 + \frac{2kd}{\sinh(2kd)}\right]\frac{\sigma k_x}{k^2} + U \tag{11}$$

$$C_y = \frac{1}{2}\left[1 + \frac{2kd}{\sinh(2kd)}\right]\frac{\sigma k_y}{k^2} + V \tag{12}$$

where $U$, $V$ = the east and north components, respectively, of depth averaged velocity;
$d$ = water depth (m);
$k$ = wave number;
$C_\sigma$, $C_\theta$ = the propagation velocities in spectral space σ-, θ-space;
S = physical processes of generation, dissipation and non-linear wave-wave interactions, the options of these terms are list in Table 1.

**Table 1.** Source Term of SWAN.

| Source Term | Scheme |
|---|---|
| Linear wind growth | Cavaleri and Malanotte-Rizzoli (1981) [20] |
| Exponential wind growth | Komen et al. (1984) [21] |
| White capping | Komen et al. (1984) [21] |
| Triad interaction | Eldeberky (1996) [22] |
| Quadruplet interaction | Hasselmann et al. (1985) [23] |
| Depth induced breaking | Battjes and Stive (1985) [24] |
| Bottom friction | Hasselmann et al. (JONSWAP) (1973) [19] |

*2.3. FETSWCM-SWAN Model*

The FETSWCM-SWAN model which is developed to compute wave-induced water level rise when coupled with SWAN [12] has been used to forecast typhoon surges and wave disasters by the Fujian Marine Forecast Center since 2016. The FETSWCM-SWAN is a two-way model coupling of FETSWCM and SWAN wherein the FETSWCM model runs first on the coupling interval using the atmospheric forcing and tidal forcing on ocean boundary but $\tau_{wx}$, $\tau_{wy}$ are set to zero. Once the time steps of coupling interval are complete, FETSWCM passes on the water levels and currents to SWAN. SWAN is driven by wind speeds, water levels, and currents computed at the vertices by FETSWCM. The water levels and ambient currents are computed in FETSWCM before being passed to SWAN, where they are used to recalculate the water depth and all related wave processes (wave propagation,

depth-induced breaking, etc.). The FETSWCM model is driven by radiation stress gradients that are computed using information from SWAN. These gradients $\tau_{wx}$, $\tau_{wy}$ are computed by:

$$\tau_{wx} = -\frac{\partial S_{xx}}{\partial x} - \frac{\partial S_{xy}}{\partial y}$$

$$\tau_{wx} = -\frac{\partial S_{yx}}{\partial x} - \frac{\partial S_{yy}}{\partial y}$$

Because of the sweeping method used by to update the wave information at the computational vertices SWAN takes much larger time steps than FETSWCM, which is diffusion- and also Courant time-step limited due to its semi-explicit formulation. The time step is 300 s in FETSWCM and 900 s in the SWAN model. Therefore, the time interval of 1800 s for the two models guarantees computational efficiency and ensures continuity of the transferred variables. We performed a series of simulations using different coupling time intervals. The processing time of the coupled model was found to be too long when the time interval was small (e.g., ≤600 s), while certain short-term tide–surge–wave interaction processes vanished when the time interval was too large (e.g., ≥3600 s). Thus, we selected the time interval of 1800 s as optimal for the requirements of this study. After each 1800-s interval, the real-time waves determined by SWAN are given to FETSWCM, and FETSWCM returns the real-time water level field and current field to SWAN. The two models perform the data exchange process as shown in Figure 1.

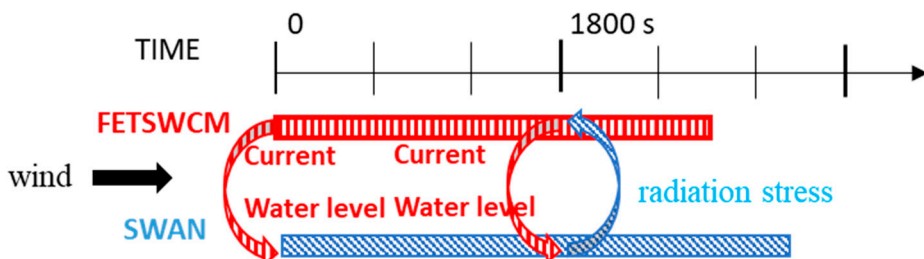

**Figure 1.** FETSWCM and SWAN coupled processing diagram.

*2.4. Model Configuration*

2.4.1. Atmospheric forcing functions

The air pressure field function is calculated considering Holland's parameter [25]:

$$P = P_e + \Delta P \, exp\left[-(\frac{r}{r_m})^{-B}\right] \tag{13}$$

where:

$P$ = the pressure at radius r;
$P_e$ = the ambient pressure;
$P_c$ = the central pressure of storm;
$\Delta P = P_e - P_c$;
$B$ is the scaling parameter of storm based on [26]:

$$B = 0.886 + 0.0177 V_{Cm} - 0.0094 \cdot lat \tag{14}$$

A typhoon model based on Holland axisymmetric model, considering the wind velocity enhancement and the cyclone structure deformation caused by the channel effect of Taiwan Strait and

the block effect of Taiwan Island is adopted [13]. The wind field model combining two correction modules with an ideal wind model is depicted as follows:

$$V = [f(V_C) + \Delta V] \times A_T + V_M \tag{15}$$

where $V_C$ is the wind speed from the Holland model [25]:

$$V_C = V_{Cm} \sqrt{\left(\frac{r_m}{r}\right)^B exp\left[1 - \left(\frac{r_m}{r}\right)^B\right]} \tag{16}$$

where $r_m$ is the radius of the maximum wind speed, $V_{Cm}$ the maximum wind speed of the typhoon which can be calculated as: $V_{Cm} = \sqrt{\frac{B\Delta p}{\rho_a e}}$,

$\Delta V$ is the additional velocity caused by topography,

$A_T$ is the attenuation factor of landing typhoon,

$V_M$ is Jelesnianski additional wind velocity [27]:

$$V_M = \begin{cases} \frac{r}{r_m + r}\vec{U}, & r \le r_m \\ \frac{r_m}{r_m + r}\vec{U}, & r > r_m \end{cases} \tag{17}$$

where $\vec{U}$ is the velocity of typhon center.

### 2.4.2. Tidal forcing functions

Seven long-period constituents (Mtm, Mf, MSf, Mm, MSm, Ssa, Sa) from NAO.99L model and major 16 short-period constituents (M2, S2, K1, O1, N2, P1, K2, Q1, M1, J1, OO1, 2N2, Mu2, Nu2, L2, T2) from NAO.99b model are used to generate tidal water level time series to drive the model at the open boundary. These models are developed by assimilating nearly 5 years of TOPEX/POSEIDON altimeter data into hydrodynamical model [28].

$$\varsigma = \sum_{i=1}^{23} f_i \cdot H_i \cdot \cos(\sigma_i \cdot t + V_i + u_i - g_i) \tag{18}$$

where $H_i$ and $g_i$ are the harmonic constants, amplitude and phase-lag of the $i$ th tidal constituent, respectively, $\sigma_i$ the angular speed, $f_i$ the nodal factor, $u_i$ the nodal angle, and $V_i$ the initial phase angle of the equilibrium constituent at Greenwich.

## 3. Validation

In 2015, five typhoons affected the Taiwan Strait area, four of which were super typhoons. Two of these super typhoons (Soudelor and Dujuan) crossed the Taiwan Strait and twice made landfall (Figure 2). The characteristics of typhoons Soudelor and Dujuan are listed in Table 2. The maximum wind speed recorded at the Suao meteorological station, which lies on the east coast of the island of Taiwan, reached 66.1 and 68.4 m/s during the passage of Soudelor and Dujuan, respectively. In Taiwan, Soudelor and Dujuan caused 6 and 2 fatalities and led to 102 and 324 people being injured, respectively. Both storms caused serious damage and the aquaculture and breakwaters along the west coast were damaged severely by high waves and flooding.

To investigate the response of the coastal water of the Taiwan Strait to the passage of these two storms, all available data were collected. Water level observations were obtained using float-type tide gauges at five tide gauge stations (Dongshan, Xiamen, Chongwu, Pingtan, and Sansha, north-to-south along the western bank of the Taiwan Strait) managed by the State Oceanic Administration of China. The measured storm surge was calculated by subtracting the predicted astronomical tide elevation

from the water level observations. Significant wave heights were obtained using four float-type wave gauges (No.2, No.3, No.3, and No.4) deployed in the Taiwan Strait and four nearshore gauges (Gulei, Douweigang, Huanqi, Beishuang) along the coast of Fujian Province. The locations of the tidal stations (solid circles) and wave buoys (open circles) are shown in Figure 2.

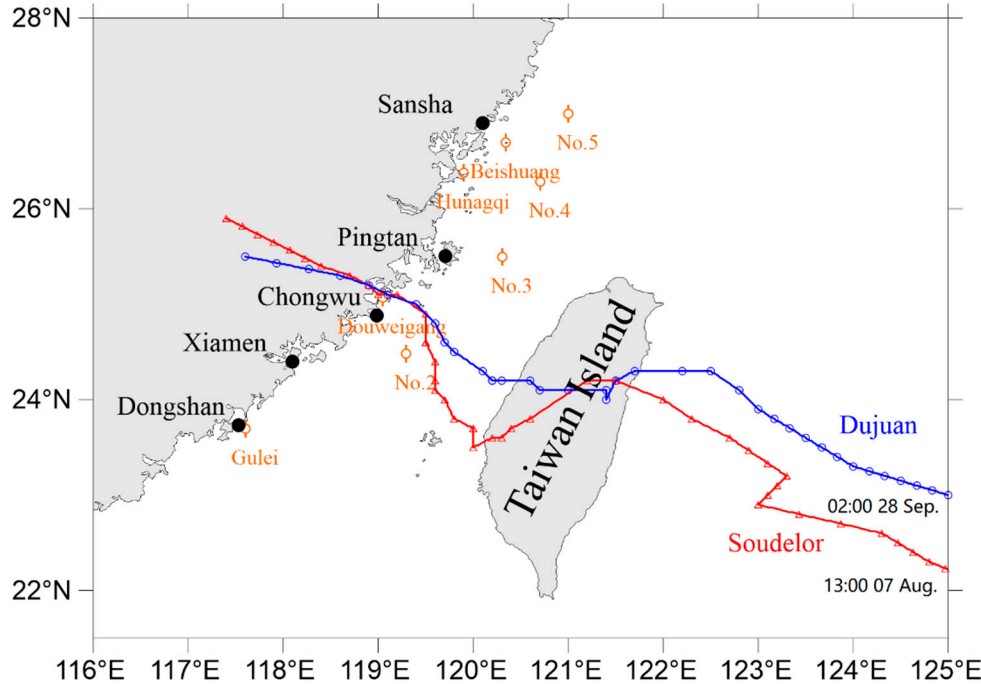

**Figure 2.** Locations of tidal stations (solid circles) and wave buoys (open circles) in the Taiwan Strait, and the tracks of typhoons Soudelor and Dujuan.

**Table 2.** Characteristics of typhoons Soudelor and Dujuan.

| Name of Typhoon | | Soudelor (2015) | Dujuan (2015) |
| --- | --- | --- | --- |
| First landfall | location | 121.5° E, 24.2° N | 121.7° E, 24.3° N |
| | time | 05:00, 8 August | 18:00, 28 September |
| | Max Wind Speed (m/s) | 45 | 48 |
| | Central Pressure (hPa) | 950 | 945 |
| Second landfall | location | 119.2° E, 25.1° N | 119.1° E, 25.1° N |
| | time | 22:00, 8 August | 09:00, 29 September |
| | Max Wind Speed (m/s) | 38 | 33 |
| | Central Pressure (hPa) | 970 | 975 |

The sea surface pressure, wind speed, and significant wave height observations acquired during the passage of both Soudelor and Dujuan are shown in Figures 3 and 4. The recorded wind speed was highest during the time between the two landfalls. Waves are driven by both wind and pressure, i.e., stronger winds and lower pressure generate higher waves. In comparison with Soudelor, Dujuan weakened more after its first landfall, which resulted in comparatively lower wave heights in the Taiwan Strait. Wave height increased rapidly as the typhoons approached and decreased as the typhoons moved away. As the typhoons crossed the area of the Taiwan Strait (11:00 on 8 August to 00:00 on 9 August for Soudelor, and 01:00–09:00 on 29 September for Dujuan), the wave height recorded at the buoys in the Taiwan Strait area peaked and then subsided as the typhoons moved away. Surface pressure decreases as a typhoon approaches and increases as a typhoon moves away. Surface wind speed, which is often nearly zero within the eye of a typhoon, increases rapidly with distance from the eye to reach its highest value at the radius of maximum wind before decreasing

steadily. Buoys No.2, No.3, and Douweigang were close to the track of both Soudelor and Dujuan (Figure 2). When the typhoons passed by, these three buoys were located in the area between the eye and the radius of maximum wind. As the typhoons approached, the surface pressure would decrease and the surface wind speed would increase. The peak surface wind would be when the buoys were at the radius of maximum wind from the eye of the storm. Wind speed would then diminish and pressure would continue to decrease as the center of the storm (point of minimum pressure) continued to move closer to the buoys. Thus, the maximum wind at these buoys would appear earlier than the minimum pressure (not the pressure peak). The other buoys were located far from the typhoon track and therefore the maximum wind speed was concurrent with the lowest pressure.

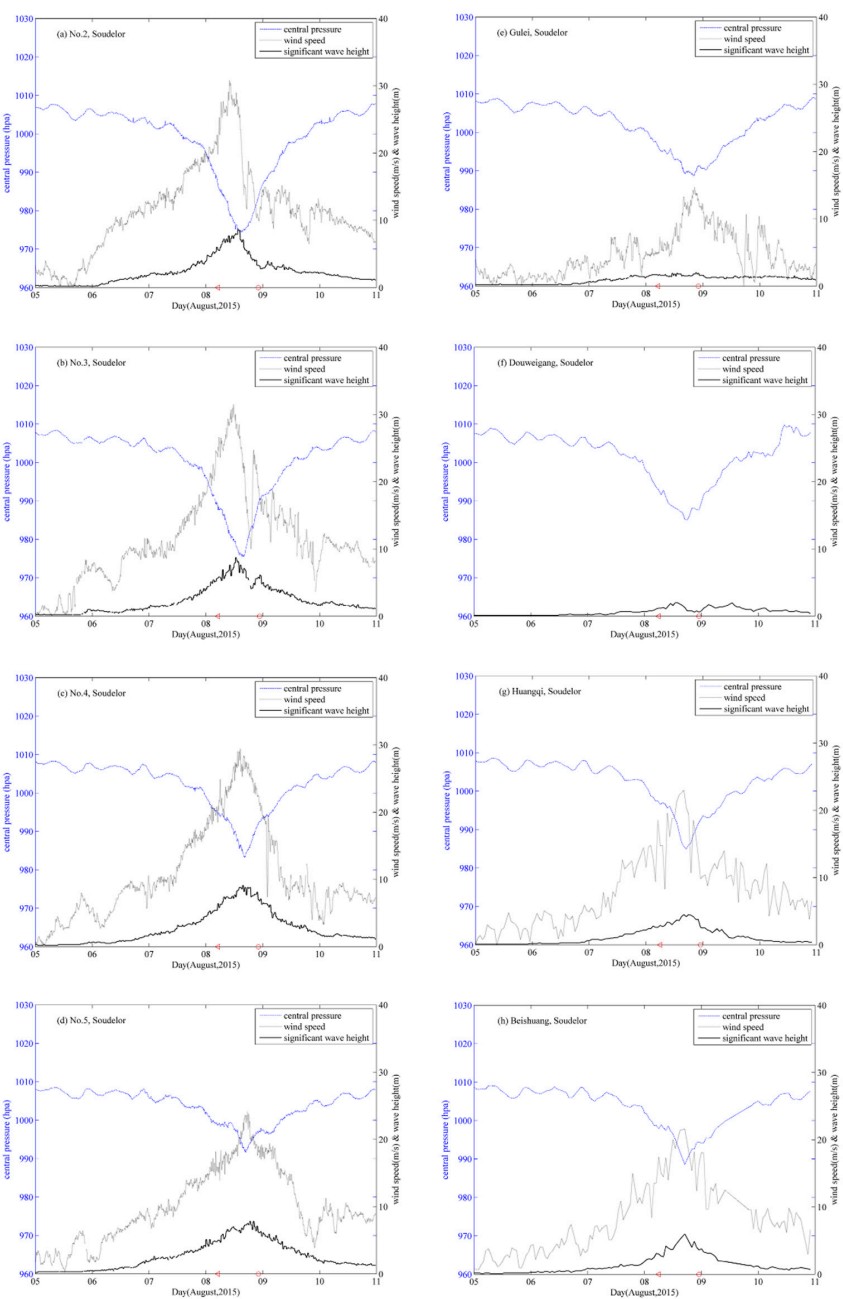

**Figure 3.** Time series of central pressure (blue dotted lines), wind speed (dashed lines), and significant wave height (solid lines) in the Taiwan Strait during the passage of Typhoon Soudelor. (**a**) Buoy No.2, (**b**) Buoy No.3, (**c**) Buoy No.4, (**d**) Buoy No.5, (**e**) Buoy Gulei, (**f**) Buoy Douweigang, (**g**) Buoy Huangqi, (**h**) Buoy Beishuang.

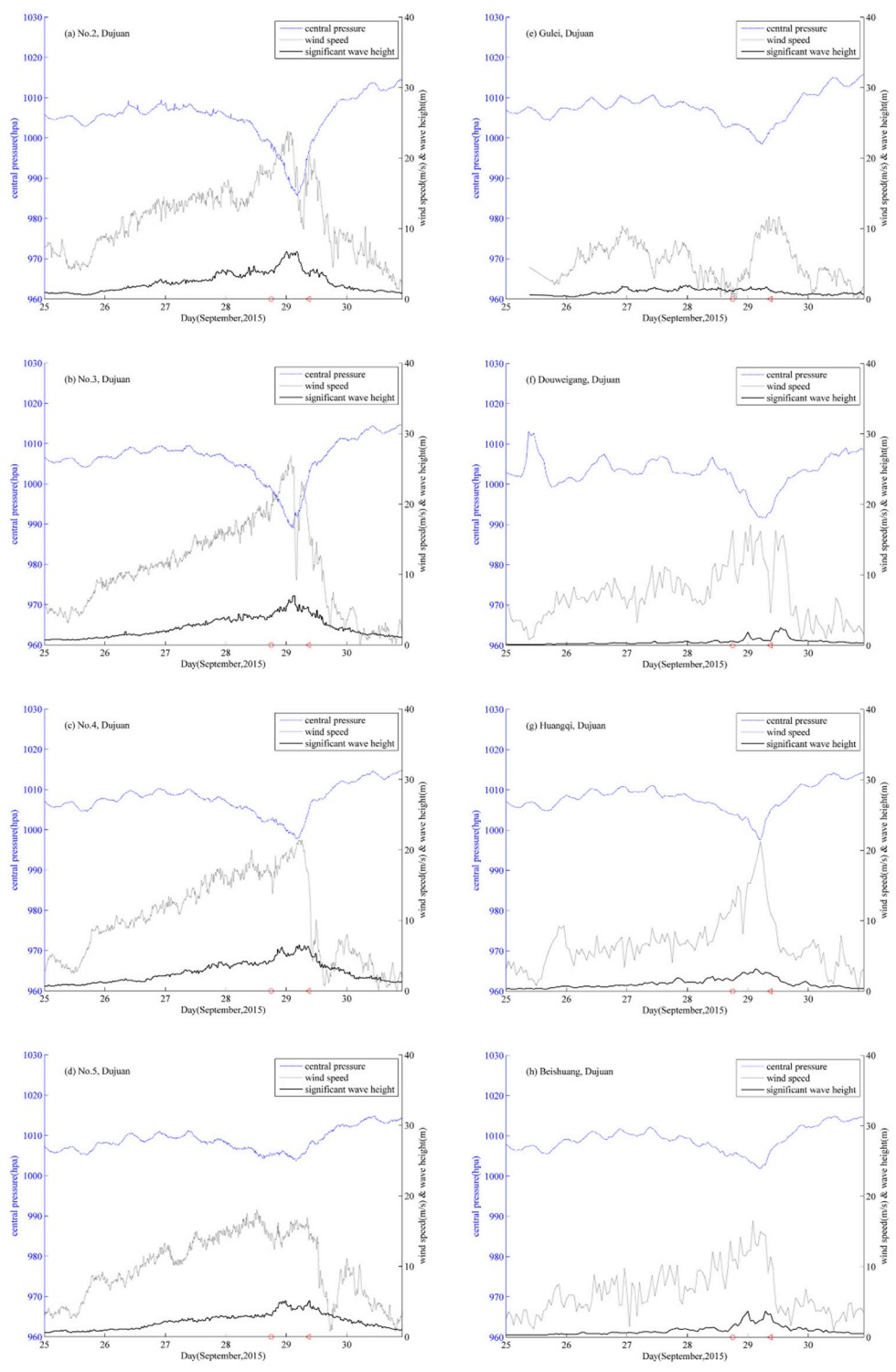

**Figure 4.** Time series of central pressure (blue dotted lines), wind speed (dashed lines), and significant wave height (solid lines) in the Taiwan Strait during the passage of Typhoon Dujuan. (**a**) Buoy No.2, (**b**) Buoy No.3, (**c**) Buoy No.4, (**d**) Buoy No.5, (**e**) Buoy Gulei, (**f**) Buoy Douweigang, (**g**) Buoy Huangqi, (**h**) Buoy Beishuang.

### 3.1. Tidal Water Level Validation

Figure 5 shows the water levels on the western bank of the Taiwan Strait during the passage of Soudelor and Dujuan, including the tidal gauge observations (black solid dots) and predictions

of water level that considered the interaction between the tide and the storm surge (red solid lines). The tidal range is <3 m in the south of the Taiwan Strait but it reaches >6 m in the north. On the western bank of the Taiwan Strait, the storm surge reached over 1 m during the passage of both Soudelor and Dujuan. The arrival of Typhoon Dujuan occurred in conjunction with the astronomical spring tide, which revealed the considerable potential danger of inundation. However, Dujuan made landfall at the time of lower tide elevations and thus severe storm surge damage was fortunately avoided.

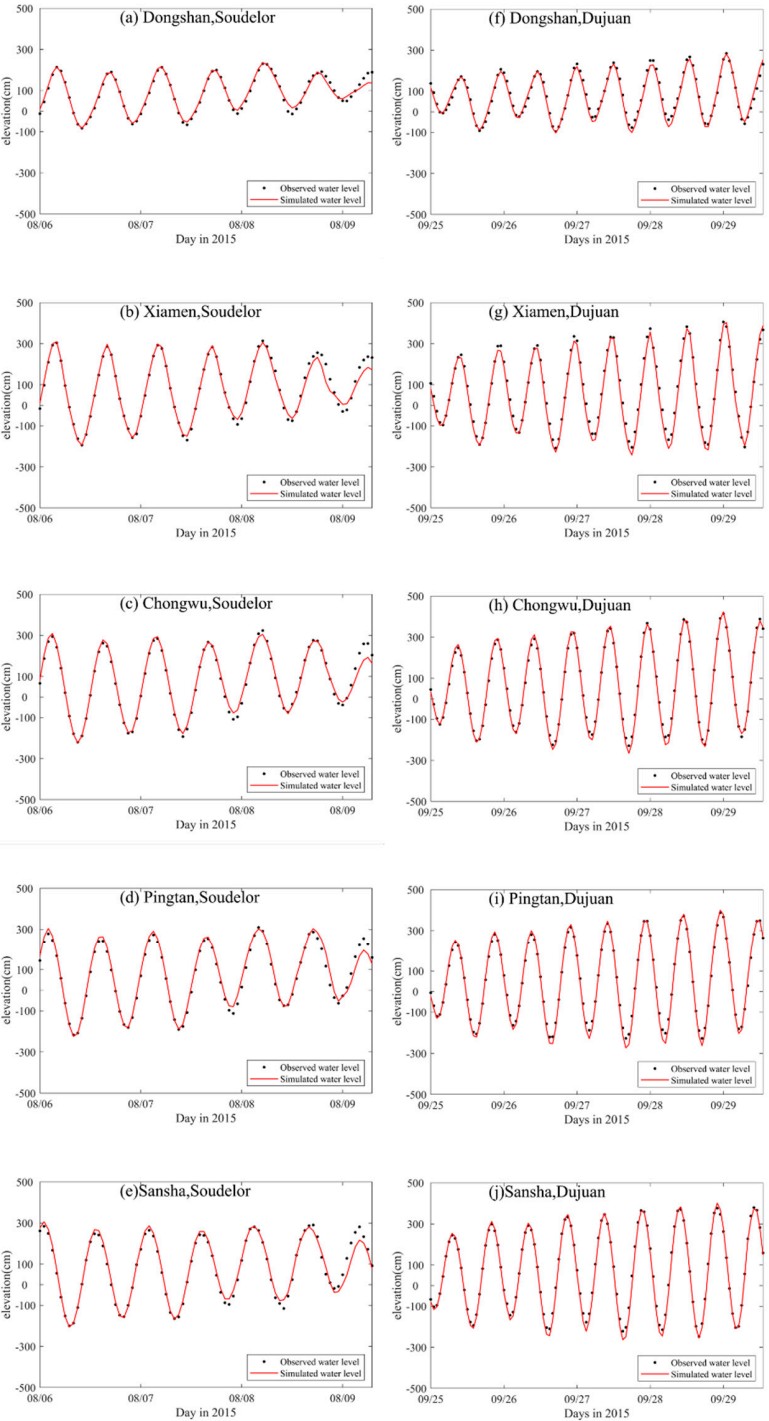

**Figure 5.** Time series of observed water level (solid dots) and predicted water level simulated with consideration of tide-surge interaction (red solid lines) of Typhoon Soudelor (**a–e**) and Typhoon Dujuan (**f–j**) on the western bank of the Taiwan Strait.

## 3.2. Significant Wave Height Validation

Time series of significant wave height observations (black solid dots) and significant wave height simulations (red solid lines) in the Taiwan Strait during the passage of typhoons Soudelor and Dujuan are shown in Figures 6 and 7, respectively.

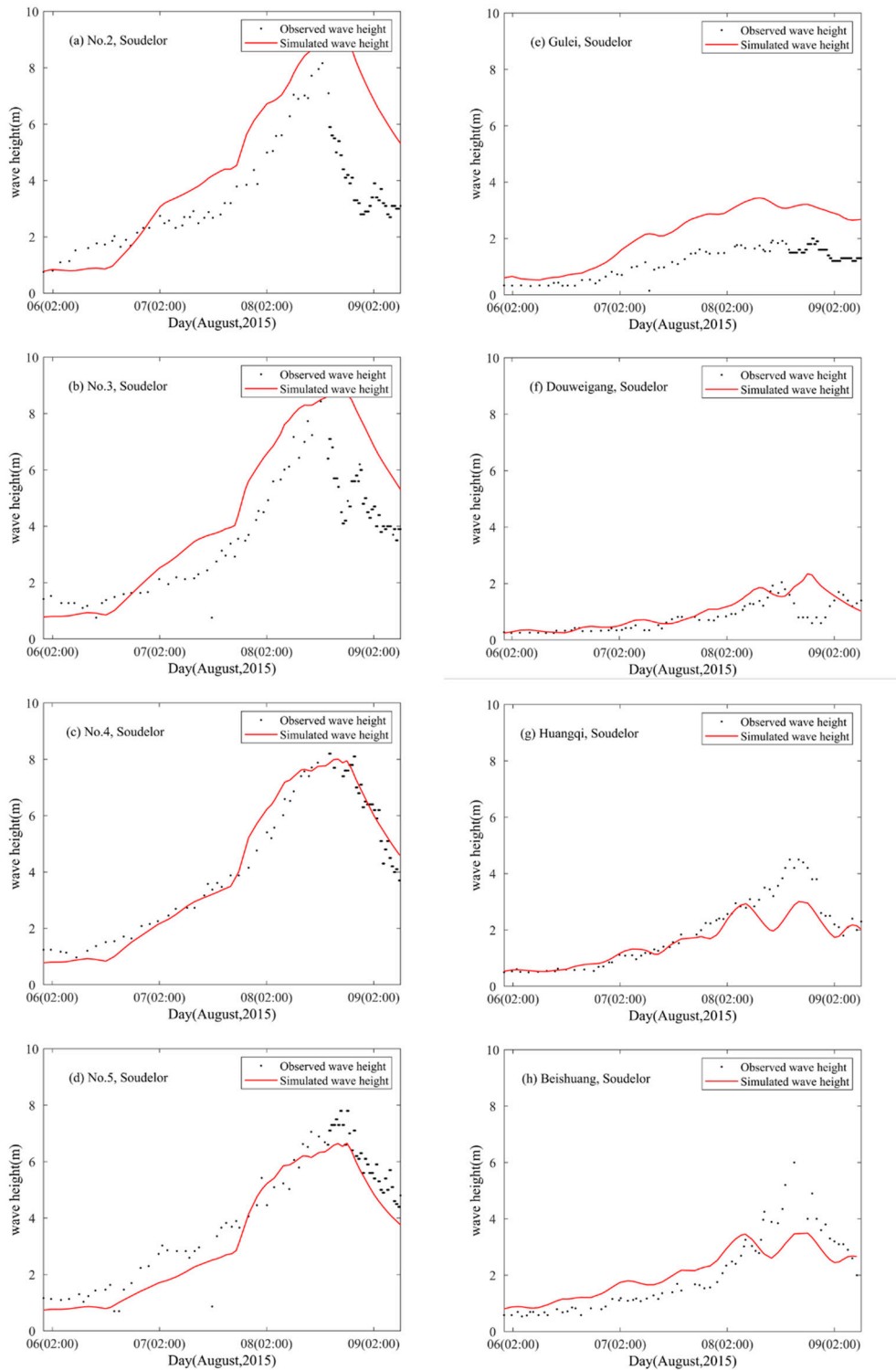

**Figure 6.** Time series of observed wave height (solid dots) and simulated wave height (red solid lines) in the Taiwan Strait during the passage of Typhoon Soudelor. (**a**) Buoy No.2, (**b**) Buoy No.3, (**c**) Buoy No.4, (**d**) Buoy No.5, (**e**) Buoy Gulei, (**f**) Buoy Douweigang, (**g**) Buoy Huangqi, (**h**) Buoy Beishuang.

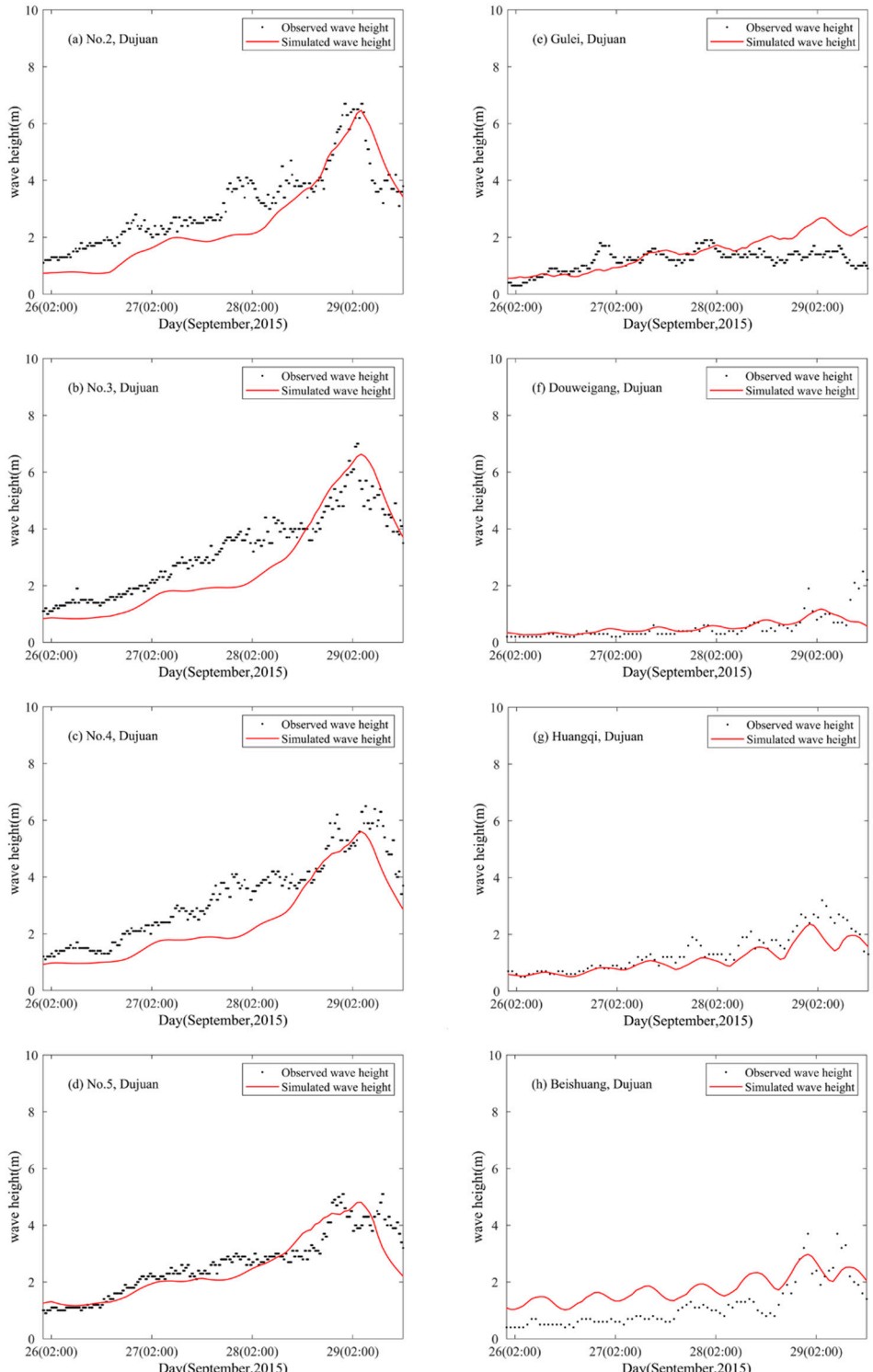

**Figure 7.** Time series of observed wave height (solid dots) and simulated wave height (red solid lines) in the Taiwan Strait during the passage of typhoons Dujuan. (**a**) Buoy No.2, (**b**) Buoy No.3, (**c**) Buoy No.4, (**d**) Buoy No.5, (**e**) Buoy Gulei, (**f**) Buoy Douweigang, (**g**) Buoy Huangqi, (**h**) Buoy Beishuang.

This study used the typhoon wind fields without consideration of the background wind, which means the wind fields were not very accurate when the typhoons were distant. The wave simulations for the periods when the typhoons were distant and after the second landfall were not very accurate. The simulation of significant wave height in the northern part of the Taiwan Strait (No. 5) is

larger, while the simulation of significant wave heights in the central part of the Taiwan Strait is more accurate. The maximum significant wave height in the Taiwan Strait was at buoy No. 2, i.e., up to 6.7 m. During 28–29 September, the measured significant wave height at the buoys in the middle of the Taiwan Strait had an obvious peak. The measured maximum significant wave height at the buoys offshore of Fujian Province was at Huangqi buoy, i.e., 3.2 m. The measured significant wave height of the offshore buoys has obvious tidal periodic oscillation, and the simulation of the significant wave height under the influence of the tidal current and water level is closer to the measured one, exhibiting obvious tidal periodic oscillation.

## 4. Discussion

### 4.1. Tide and Storm Surge Interaction

Two sets of numerical experiments were conducted to assess the model performance and to analyze the mechanism of tide storm surge interaction: (a) Full run: The model was driven by both the tidal forcing at the ocean boundary and the atmospheric forcing. The resultant water level from this model run is the storm tide. (b) Storm-only run: Only the blended atmospheric forcing was used to drive the model, while the tidal forcing was turned off. The resultant water level from this model run is called pure storm surge. (c) Tide-only run: Only the tidal forcing was included. The resultant water level is the pure astronomical tide level.

In the Taiwan Strait, large tidal range and severe storm surge are observed and tide–surge interaction is remarkable, especially near the western bank. The storm surge observations always show tidal oscillation because of the tide–surge interaction [6,29]. According to experiments conducted by Zhang et al. [6], nonlinear bottom friction is a major factor that should be considered when predicting these oscillations. In addition, the presence of the island of Taiwan enhances tide–surge interactions.

During the passage of both Soudelor and Dujuan, the storm surge showed oscillation within the tidal period that was inversely proportional to the tide elevation, i.e., low at high tide and high at low tide. It confirms that tide–surge interaction is inevitable in coastal areas with shallow water depth and large tides. The predicted water level matched the observations well, especially the predictions that considered tide–surge interaction. Without consideration of tide–surge interaction, the predictions were higher at high tide and lower at low tide, which resulted in overprediction of the tidal range.

The peak water levels at Xiamen and Dongshan occurred earlier than the astronomical high tide owing to tide–surge interaction. The tides from the northern and southern entrances of the Taiwan Strait mix in the district between Dongshan and the Penghu shallows [1,11]. In the northern part of the Taiwan Strait, tidal wave propagation was accelerated because the water depth was increased by the storm surge, which caused the peak to arrive earlier. This type of pre-arrival was also evident at Xiamen and Dongshan.

Discrete Fourier transform (DFT) analysis, applied to both the astronomical tide and the interaction residual (difference between the predicted water level simulated with and without consideration of tide–surge interaction), showed that tide–surge interaction led to oscillation within the tidal period. Oscillation around the period of the M2 tidal constituent was most remarkable (Figures 8 and 9). Table 3 shows the DFT results for typhoons Soudelor and Dujuan. In comparison with Typhoon Dujuan, tide–surge interaction was stronger during the passage of Typhoon Soudelor because of typhoon intensity.

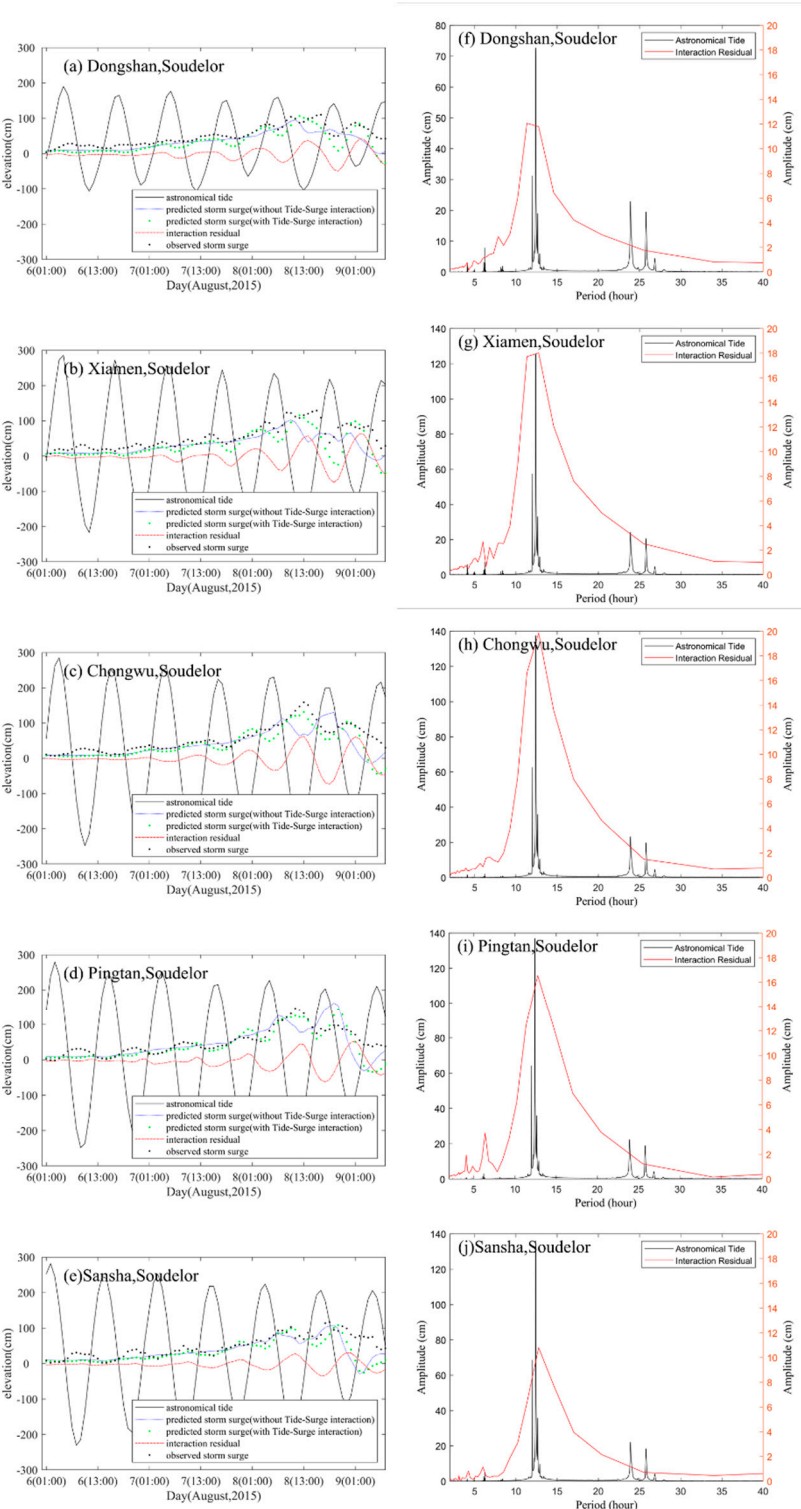

**Figure 8.** Time series of astronomical tide water level (solid lines), predicted water level without consideration of tide-surge interaction (dotted lines), predicted water level simulated with consideration of tide-surge interaction (green solid dots), and interaction residual (red dashed lines) during the passage of Soudelor (**a–e**), together with discrete Fourier transform of astronomical tide (black solid lines) and interaction residual (red solid lines) of each station (**f–j**).

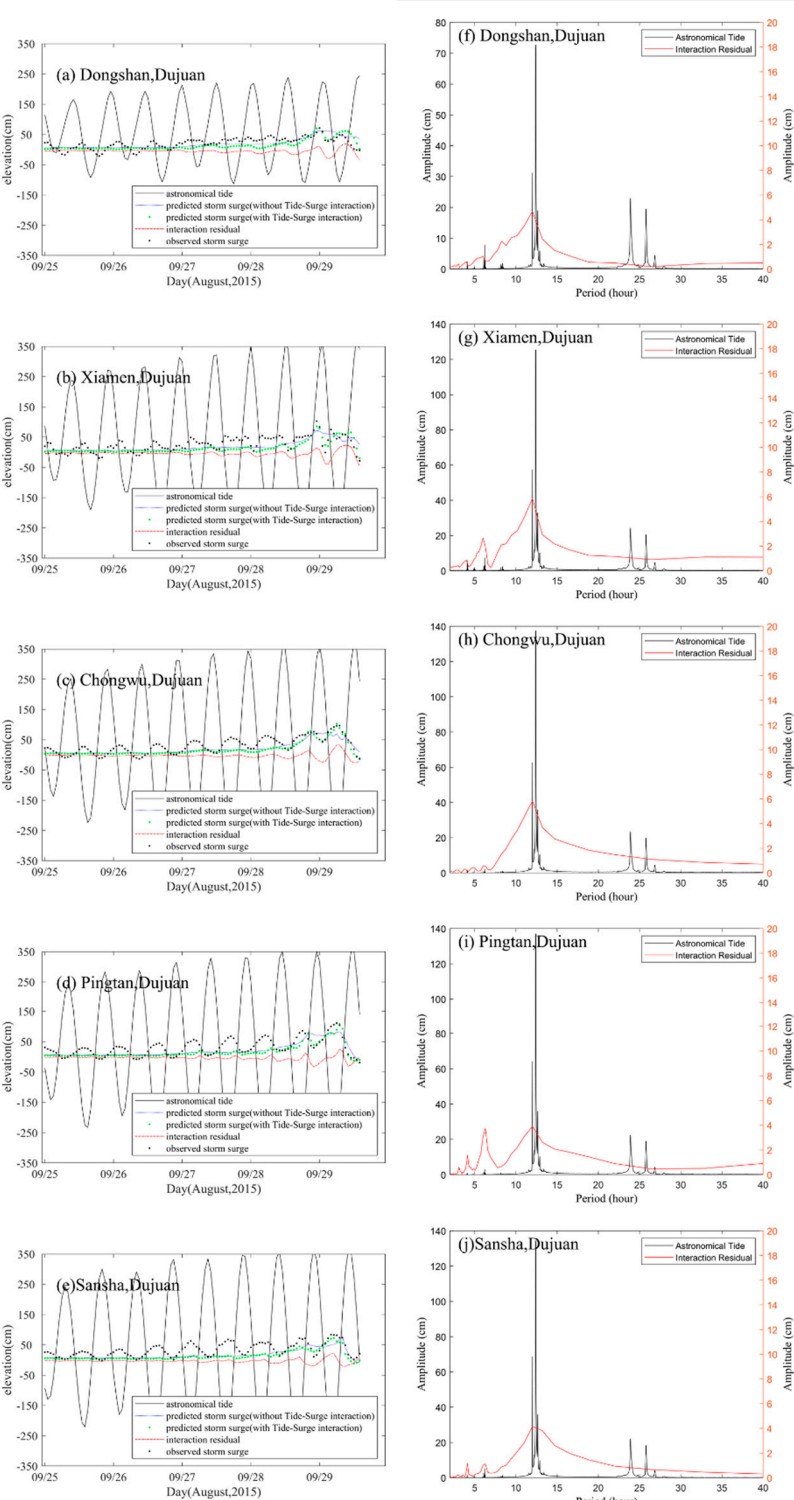

**Figure 9.** Time series of astronomical tide water level (solid lines), predicted water level without consideration of tide-surge interaction (dotted lines), predicted water level simulated with consideration of tide-surge interaction (green solid dots), and interaction residual (red dashed lines) during the passage of Dujuan (**a**–**e**), together with discrete Fourier transform of astronomical tide (black solid lines) and interaction residual (red solid lines) of each station (**f**–**j**).

**Table 3.** Discrete Fourier transform results for typhoons Soudelor and Dujuan.

| | Typhon Soudelor | | | Typhon Dujuan | | |
|---|---|---|---|---|---|---|
| | M2(cm) | Interaction Residual (cm) | % | M2 (cm) | Interaction Residual (cm) | % |
| Sansha | 134.7 | 10.8 | 8.0 | 134.7 | 4.2 | 3.1 |
| Pingtan | 137.1 | 16.6 | 12.1 | 137.1 | 3.9 | 2.9 |
| Chongwu | 137.6 | 19.9 | | | 14.4 | 5.8 | 4.2 |
| Xiamen | 124.5 | 18.0 | 14.5 | 125.5 | 5.9 | 4.7 |
| Dongshan | 72.7 | 11.8 | 16.2 | 72.7 | 4.6 | 6.3 |
| average | | | 13.0 | | | 4.2 |

### 4.2. Wave and Tide Surge Interaction

To determine the relative effects of the current and water level, three sets of numerical experiments were conducted to assess the model performance and to analyze the mechanism of wave water level interaction: (a) Full Coupled run (Run-Full): SWAN was driven by atmospheric forcing, current and water level to simulate wave height ($H_{Full}$). The results showed that both the current and water level affect waves during a typhoon process. (b) Current coupled only run (Run-CO): SWAN was driven by atmospheric forcing and current to simulate wave height ($H_{CO}$). (c) Water level coupled only run (Run-WO): SWAN was driven by atmospheric forcing and water level to simulate wave height ($H_{WO}$). (d) No Coupled run (Run-NC): neither current nor water level is provided, SWAN is driven by only atmospheric forcing to simulate wave height ($H_{NC}$). The wave height variations were determined by subtracting the wave heights of the No Coupled run from the other cases.

In nearshore areas, tides can have significant effect on short waves. First, wave heights are modulated along a tidal cycle because tide-induced water level variations shift the cross-shore position of the surf zone [30]. Second, in coastal zones subjected to strong tidal currents (e.g., tidal inlets), tidal currents can substantially affect the wave field [31]. A tidal wave is a progressive wave. During the flood phase, the direction of wave propagation is consistent with the current direction, which is not conducive to wave growth; thus, the tidal current flattens the wave amplitude and reduces the wave height. However, during the ebb phase, as and when the current increases, wave steepness increases with an increase in wave height together with a decrease in wavelength.

In the Taiwan Strait area, interaction between tides and waves was evident. According to the simulations (a–d of Figures 10–13), wave height oscillated within the tidal period and this oscillation was most obvious in the nearshore area. Waves break with shoaling and wave height was affected by both the current and the water depth. Wave height was affected markedly by the storm surge current, which was opposite to the direction of wave propagation at the Gulei and Douweigang buoys during the passage of Typhoon Dujuan. The wave height variations were larger in the nearshore areas where the wave height was low. The maximum wave height variation was 0.98 m at the Huangqi buoy during the passage of Typhoon Dujuan; the oscillation amplitude was up to 0.40 m, which is comparable with the mean value. In the northwest of the Taiwan Strait (i.e., at the Huangqi and Beishuang buoys), where the tidal range is comparable with water depth, the wave height was affected mostly by water level. The high (low) tide was accompanied by higher (lower) wave height.

Based on the DFT method (right-hand column of Figures 9–12), the numerical simulation results indicated that waves were affected mostly by the current in the middle of the Taiwan Strait, where water depth is >20 m. The contribution of current to wave–surge interaction is approximately 0.04 m, whereas the contribution of water level is <0.01 m. Conversely, it was found that the water level was more important than the current in coastal areas when interaction occurred. The contribution of current to wave–surge interaction is up to 0.18 m at Beishuang station, whereas the contribution of water level is 0.08 m at the Gulei and Beishuang stations.

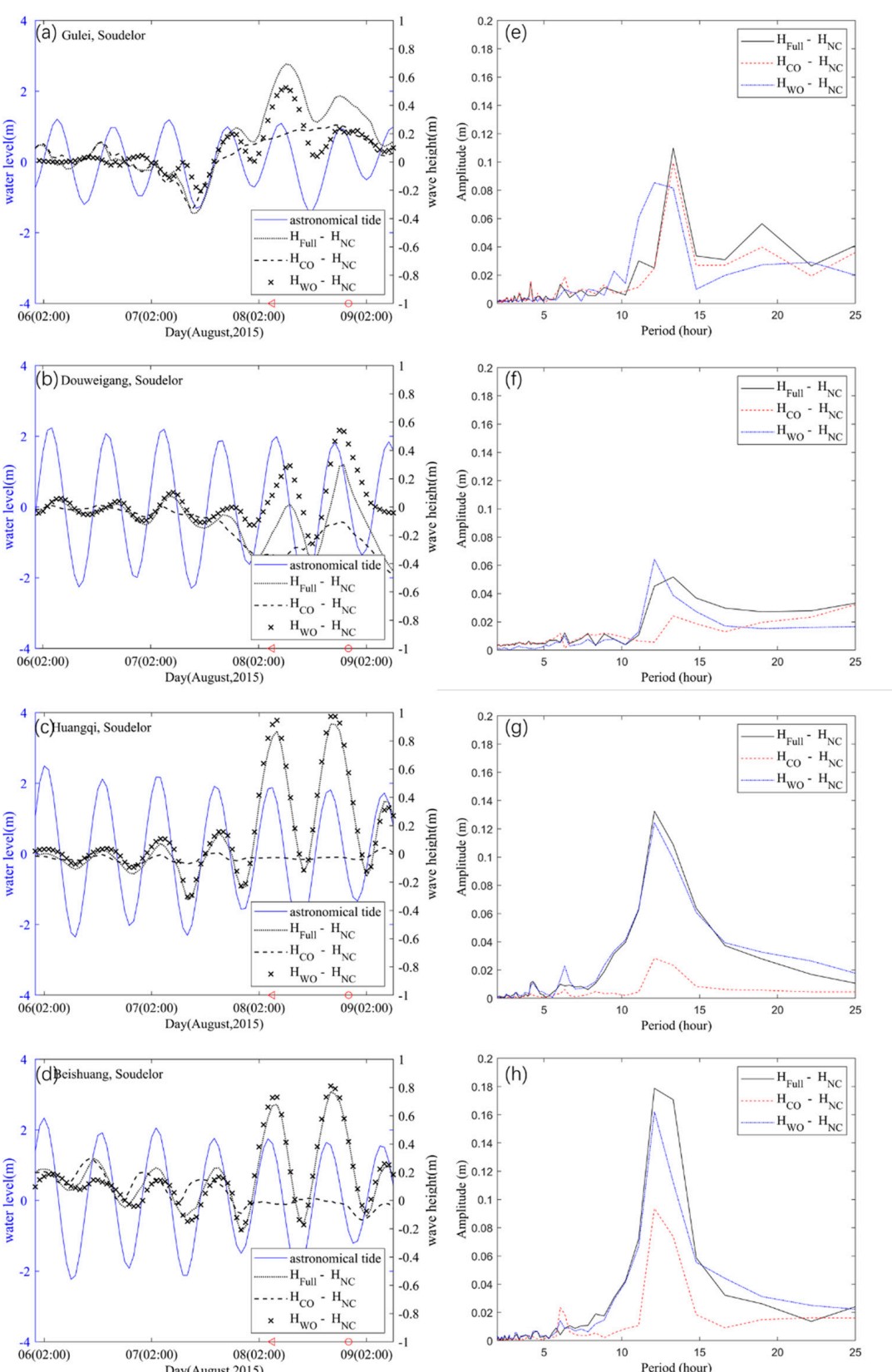

**Figure 10.** Wave-current-water level interaction during the passage of Typhon Soudelor at coastal stations (**a**–**d**), together with the results of the discrete Fourier transform (**e**–**h**).

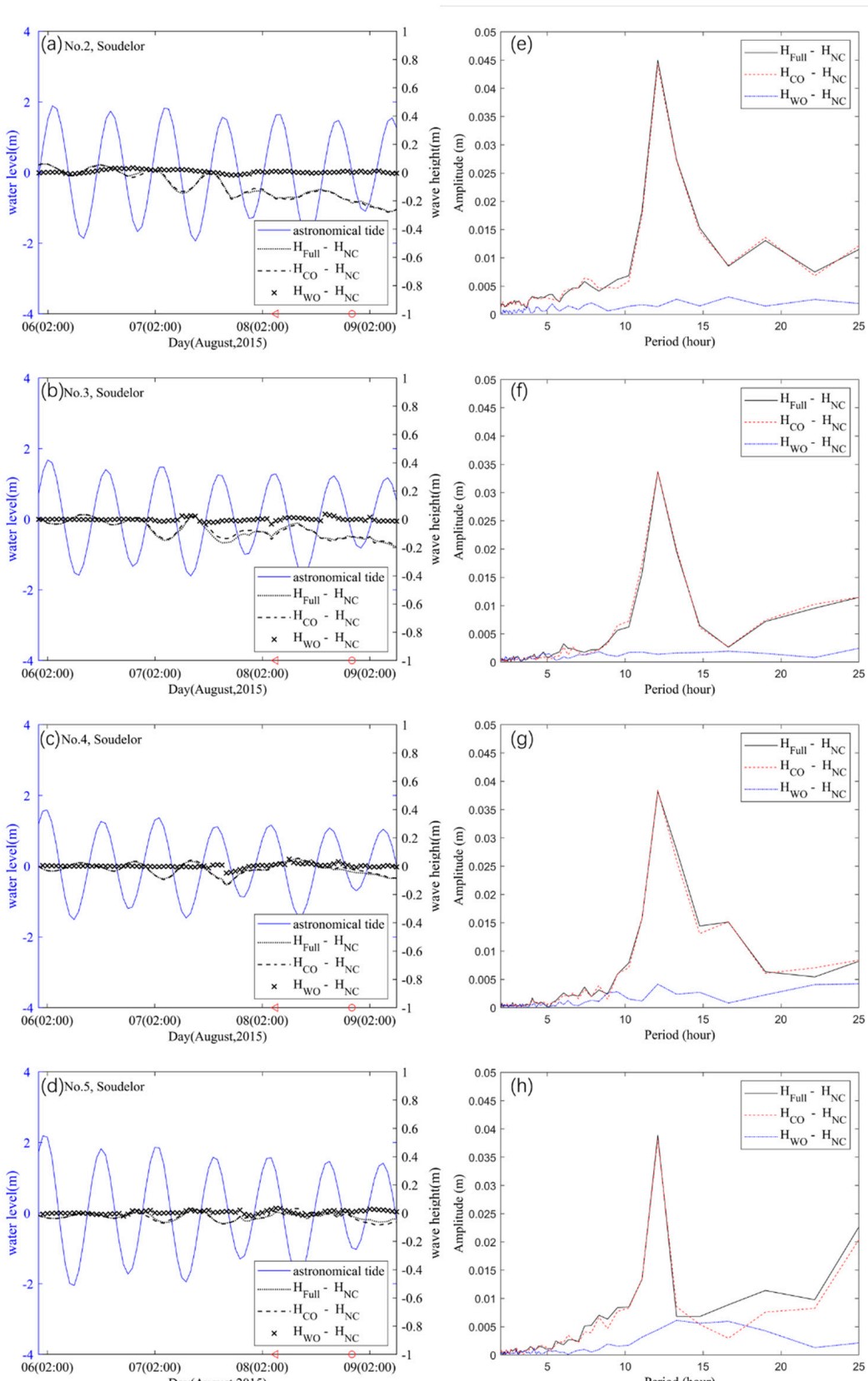

**Figure 11.** Wave-current-water level interaction during the passage of Typhon Soudelor at stations far from the coast (**a**–**d**), together with the results of the discrete Fourier transform (**e**–**h**).

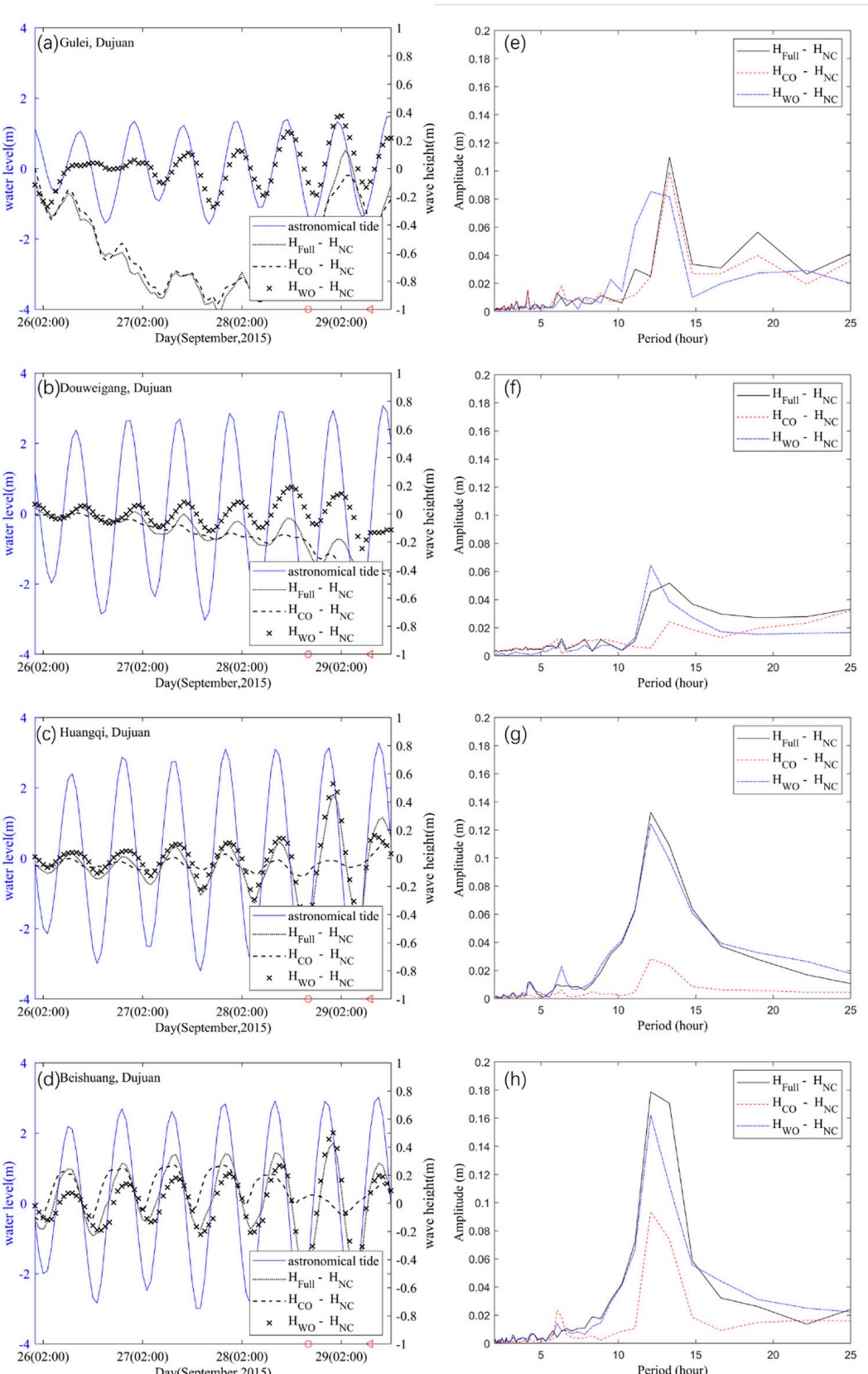

**Figure 12.** Wave-current-water level interaction during the passage of Typhon Dujuan at coastal stations (**a**–**d**), together with the results of the discrete Fourier transform (**e**–**h**).

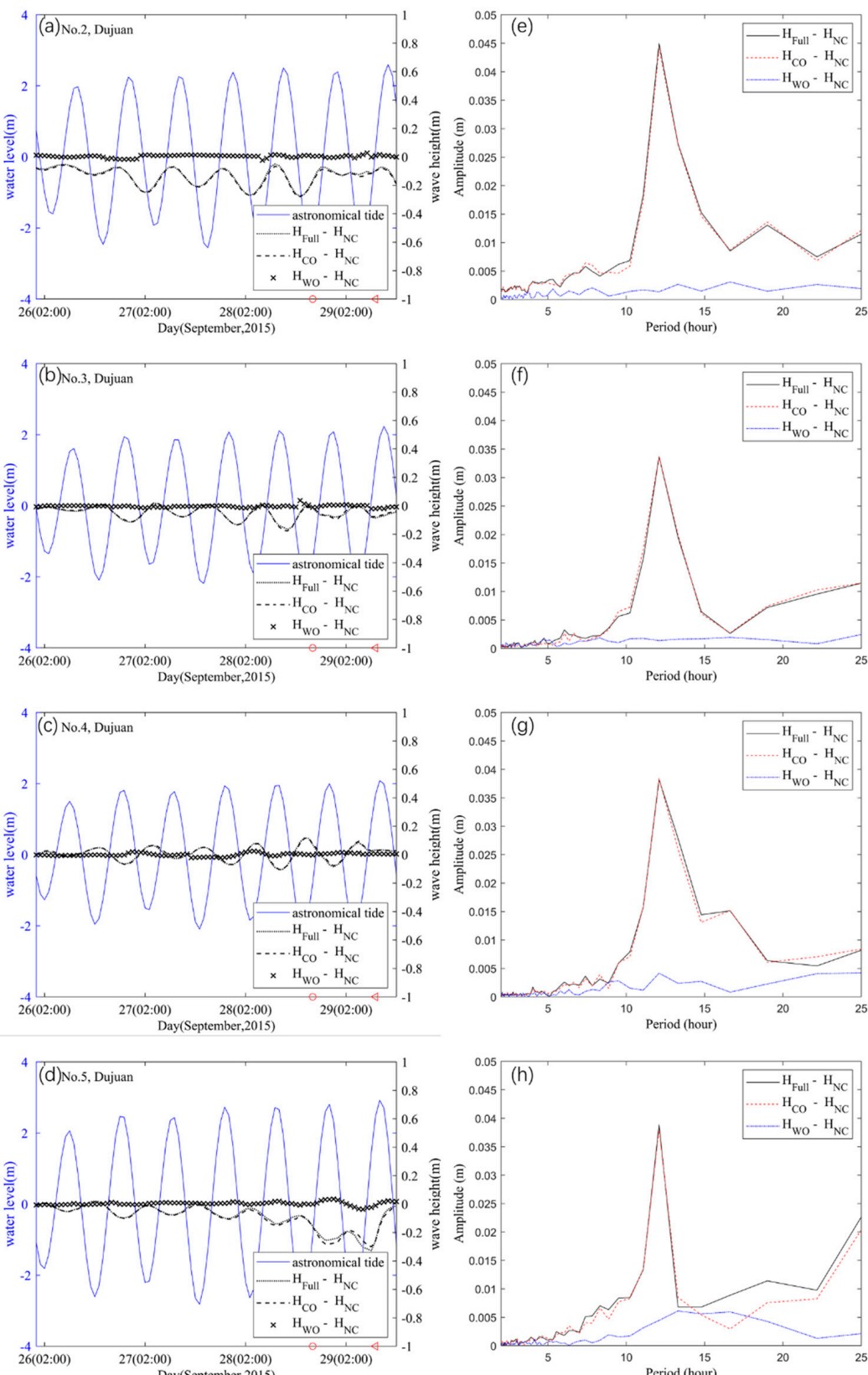

**Figure 13.** Wave-current-water level interaction during the passage of Typhon Dujuan at stations far from the coast (**a**–**d**), together with the results of the discrete Fourier transform (**e**–**h**).

### 4.3. Wave Setup in the Taiwan Strait during Soudelor and Dujuan

Wave setup is the change in mean water level due to the presence of breaking waves. In this study, wave setup was taken as the water level rise driven by the spatial variation of wave radiation stress, which is also called the wave radiation stress gradient. The wave radiation stress is the depth-integrated and phase-averaged excess momentum flux caused by waves, which is exerted on the mean flow. The variation of the radiation stress also induces change in the mean flow [32].

According to Chen et al. [16], the wave setup on the northeast coast of Taiwan could reach 1.0 m because of offshore giant (>20 m) waves. Huang et al. [9] studied wave-induced surge in Tampa Bay (Florida), while Bertin et al. [10] studied the wave setup in the Bay of Biscay; their results showed that the regional maximum wave setup could be >0.3 m. These studies found that wave setup is considerable in storm surge modeling when large wave heights occur in a region with a steep sea bottom slope. The Taiwan Strait lies on the continental shelf of the East China Sea with average depth of 60 m; therefore, there has been little study of wave-induced surge in this area.

Figures 14 and 15 shows the wave setup simulation for the western bank of the Taiwan Strait during the passage of both Soudelor and Dujuan. The wave setup was enhanced as the typhoons approached and reached its highest value when the typhoons made landfall. Although the spatial distribution of the wave setup was similar in each case, wave action was stronger and the wave setup was greater during the passage of Typhoon Soudelor.

In the deep ocean, the radiation stress gradient can be neglected because of the depth integration. However, in nearshore areas, waves break with the rapidly changing bathymetry and the radiation stress varies considerably, which means the wave setup generated by the radiation stress gradient is obvious in such areas. Wave radiation stress reaches its maximum value before the waves break and it then decreases afterward. The radiation stress gradient is directed onshore in the zone of breaking waves and directed offshore in the area of wave growth. Figure 16 shows maximum wave setup of all simulation area during the Typhon Soudelor and Dujuan. The maximum of wave set-up is about 15 cm and the minimum of wave set-up is about −8 cm during the typhon process. Consequently, wave setup, which was found positive (negative) in the nearshore (offshore) area, was over 15 (−8) cm in this study.

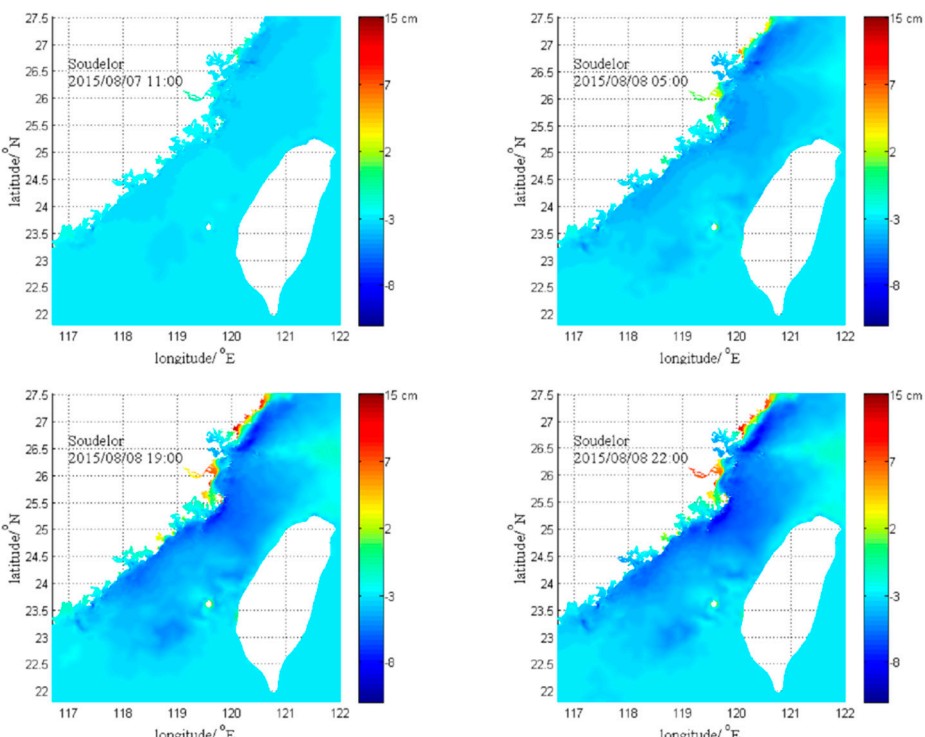

**Figure 14.** Wave setup in the Taiwan Strait during the passage of Soudelor.

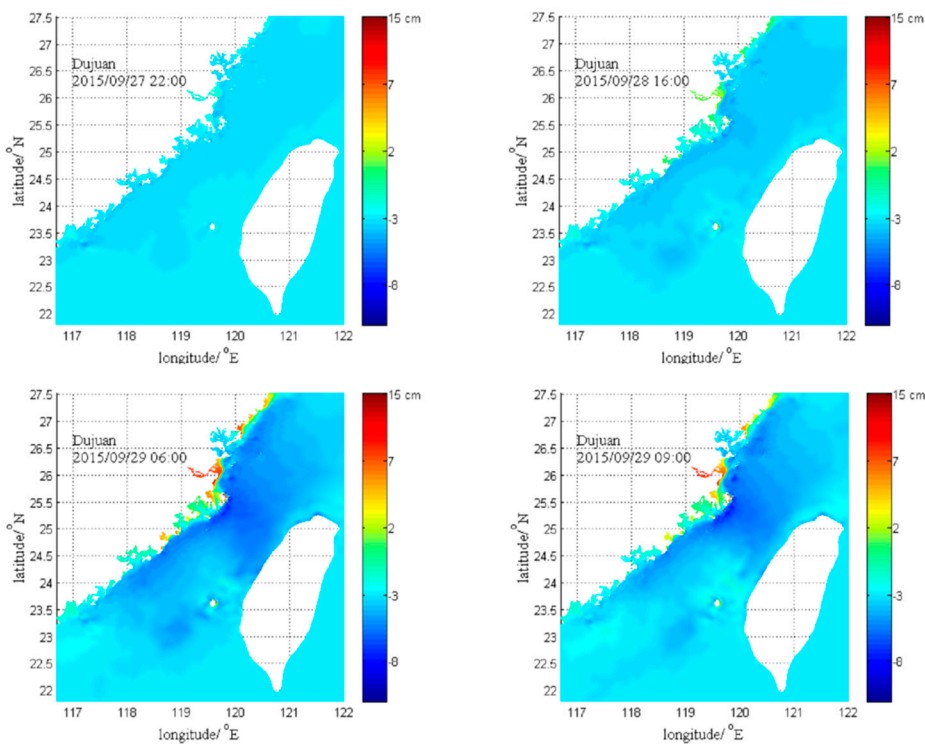

**Figure 15.** Wave setup in the Taiwan Strait during the passage of Dujuan.

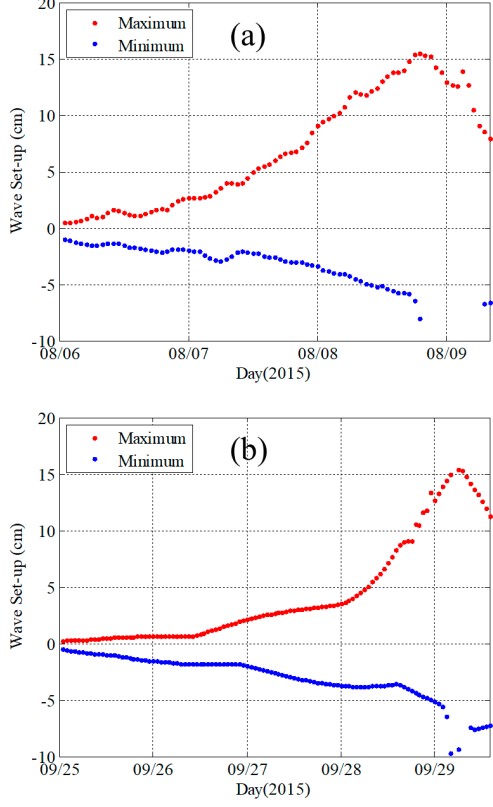

**Figure 16.** Maximum (red dot) and minimum (blue dot) wave setup during the Typhon Soudelor (**a**) and Dujuan (**b**).

It is notable that the wave setup in Sanduao Bay (26.3–26.9° N, 119.5–120° E), which was very low in comparison with the adjacent area, could also be explained by the local topography (Figure 17). The effect of shoaling is diminished at the mouth of Sanduao Bay because of a trend of deepening. Therefore, wave breaking in this zone was weakened and the wave radiation stress gradient was low, which resulted in a small wave setup. The principal finding of this study was that the distribution of wave setup was determined primarily by local topography. The wave setup also oscillated within tidal period because the waves were affected by the tide. In this study, the oscillation of the wave setup reached 3 cm at most. In the nearshore area, wave breaking was weakened at high tide, which reduced the radiation stress gradient and decreased the wave setup. Conversely, the wave setup was increased during low tide.

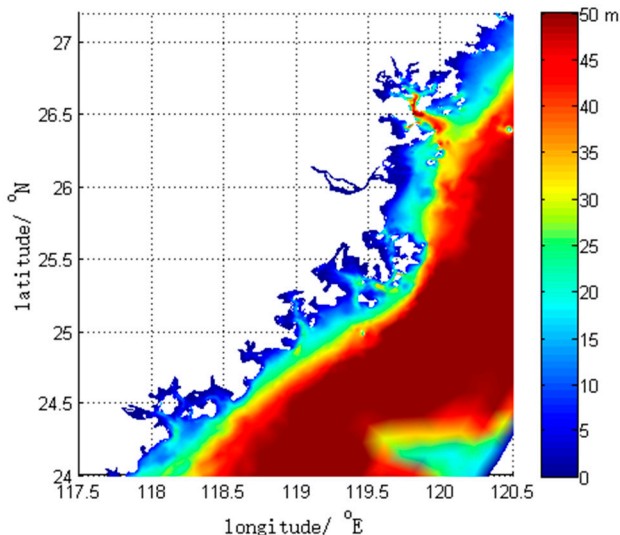

**Figure 17.** Water depth on the western bank of the Taiwan Strait.

## 5. Conclusions

In 2015, five typhoons affected the area of the Taiwan Strait, four of which were super typhoons. This study focused on the response of the waters of the Taiwan Strait to the passage of typhoons Soudelor (2015) and Dujuan (2015), which were the two strongest typhoons to affect the Taiwan Strait area in 2015. Both Soudelor and Dujuan made landfall twice and caused severe damage in China. We analyzed observations from buoys and tide stations acquired during the passage of the two typhoons, as well as the results of numerical modeling simulations.

Analysis revealed that prominent resonant coupling between the astronomical tide and the storm surge resulted in a tidal period oscillation on the storm surge and reduced tidal range. The high tide arrived early because the tide wave was accelerated by the rise of the water level attributable to the storm surge. It confirms that tide–surge interaction is inevitable in coastal areas with shallow water depth and large tides. The predicted water level matched the observations well, especially the predictions that considered tide–surge interaction.

Wave observations and modeling results showed that wave height oscillated within the tidal period. Waves are affected by both currents and water level. Wave heights are modulated along a tidal cycle because of tide-induced water level variations shift the cross-shore position of the surf zone. In coastal zones subjected to strong tidal currents such as tidal inlets, tidal currents can substantially affect the wave field. Tidal current flattens the wave amplitude and reduces the wave height. However, during this ebb phase, as and when currents increase, the wave steepness increases with the increase in wave height together with the decrease in wavelength. This study found that waves in the middle of the Taiwan Strait were affected primarily by the current, whereas water level had the greatest effect when the water level was comparable with the water depth in coastal area.

Wave setup, which is the water level rise driven by typhoon waves, was also simulated for the passage of typhoons Soudelor and Dujuan. The wave setup was enhanced as the typhoons approached and reached its highest value when the typhoons made landfall. It was found that wave setup also oscillated within the tidal period. The wave setup, which was found to be positive (negative) in the nearshore (offshore) area, was over 15 (−8) cm in this study. According to the simulation results, local terrain is the most important influencing factor of wave setup distribution.

**Author Contributions:** Conceptualization, S.S. and L.Z.; Methodology, L.Z.; Software, L.Z.; Validation, L.Z. and F.Z.; Formal Analysis, S.S.; Investigation, L.Z.; Resources, Y.X. and S.S.; Data Curation, Y.X.; Writing—Original Draft Preparation, L.Z. and F.Z.; Writing—Review & Editing, S.S., F.Z., and Y.X.; Visualization, L.Z. and F.Z.; Supervision, S.S.; Project Administration, S.S. and Y.X..; Funding Acquisition, S.S. All authors have read and agreed to the published version of the manuscript.

**Funding:** This research was funded by National Key R&D Program of China grant number No. 2017YFC1404804 and 2016YFC1401104 and Marine Economic Development Subsidy Project of Fujian, China grant number No. ZHHY-2019-2.

**Conflicts of Interest:** The authors declare no conflict of interest.

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
