# Peer review of "Tide-Surge-Wave Interaction in the Taiwan Strait during Typhoons Soudelor (2015) and Dujuan (2015)"

_applsci, doi:10.3390/app10207382_

Round 1
Reviewer 1 Report
The manuscript is dealing with a very important issue and is combine extended in-situ data and modeling results. But a better presentation and more clarification is needed, before to be ready for publication.
Also, the authors should add more references especially about the models used and their coupling procedure or extend the relevant sections.
Comments
Lines 87-89: A better documentation should be added for the parameter το
Line 96: equation (4) should be written in clear format
Line 99: as above for the equations (6)
Line 101: Is the value of Cd=0.0015 a result of model calibration?
Lines 103-104: More information should be presented for the tidal input of the model.
Line 112: More information/documentation for the FETSWCM and FETSWCM-SWAN models should be presented.
Lines 118-119: Is the time interval of 1800s a result of the calibration of the coupled model?
Figure 1: A more detail description of the coupling prosses should be given.
In sections 2.1, 2.2 and 2.3 some information presented for a series of models, i.e FETSCM, SWAN, FETSWCM and FETSWCM-SWAN, without a clear description of the whole coupling prosses and the application procedure of the models to the area under investigation.
Also, in the section 3 “validation”, and the sub-sections, more information needed for the model (or models) application and their forcing data. The plot quality should be improved.
Lines 137-145: The names of the stations in the text, do not correspond to those in figure 2. I suppose that eg. Gl is Gulei, and Dwg is Douweigang, etc. Is that correct?
Line 175, “Significant …. of sea state”: what is the meaning of this statement?
Lines 176-180: These are information about the two typhoons and the data collected during their passage. Therefore, I proposed to transfer this paragraph to the introduction of the section no. 3, before section 3.1.
Lines 181-192: These are information about the two typhoons and the data collected during their passage. Therefore, I proposed to transfer this paragraph to the introduction of the section no. 3, before section 3.1.
Lines 201-202, “The wave …. Dujuan”: This statement should be explained more
Figure 5. Time series from the Dujuan measurements and simulations only!
The quality of the figure 6 is very poor, not acceptable for presentation.
Lines 224-227, “The predicted … range”: How these statements are supported?
Lines 265-267: How these statements are supported?
Lines 268-274: How these statements are supported?
Lines 275-284: The presentation in this paragraph emphasize the need of a better explanation of the modeling procedure and the coupling of the models used. See also, the comments for figure 1, above.
Fig. 13: not in good quality for publication.
Lines 318-322: The results presented in fig. 13, showing a very low wave set-up (smaller that 8 cm), are not in agreement with above statements of lines 311-317.
Lines 328-329, “Consequently …. this study”: This statement is not supported by the results shown.
Fig. 14: What is the need of this figure?
Conclusions are too sort. Should be extended to include the main findings of this manuscript.
Reviewer 2 Report
This study shows how water levels and wave heights are affected by tide-surge-wave interactions during typhoon events. Although no new general scientific findings are presented, this regional study provides additional evidence for the past studies as well as useful information for scientists who focus on the area. While the main messages are understandable, the manuscript can improve its readability. For example, some results in Discussion should be in the previous sections. Also, in-depth discussions would improve paper quality.
Specific comments:
Line42- : Please use citation numbers in order
Line81: Explain the wind model a bit more
Equations(4)&(6): It seems to be a technical error – vectors(?) changed to other symbols
Line104: correct to NAO.99b
Figure2: What do the solid blue circles and solid red triangles mean? Corresponding times on typhoon tracks would be helpful - e.g., show time/date for both ends of the tracks and indicate increment time using the circles/triangles
Line156&164: Aren’t they open dots?
Line198: Why was the background wind information not used?
Line201-202: Is this statement based on qualitative analysis? Please show statistical results for quantitative analysis. Also, what made the oscillatory patterns?
Line221-227: Is any evidence for these statements presented in the result (3. Validation) section?
Figure7-12: It would be nice to overlay with observed data.
Line261-263: Again, what is the mechanism for the oscillation?
Figure9-12: Use more descriptive case names such as “case2-current only”.
Reviewer 3 Report
The Authors come up with a manuscript with a highly detailed work on typhoon action related with tidal, surge and wave and their mutual interaction. Indeed, the study comes at hand when dealing with the storm evolution in seas with short fetches and somewhat enclosed geographical boundaries. The work is correctly structured, well paced and addressing standards both in quality of presentation an English grammar and structure. The paper is worth of publishing in the context of Applied Sciences Journal. Some tips that should be amended before proceed to a final recommendation:
- Some typos in Eqn (6), line 98 and on.
- The captions, axis labels and general layout in figures should be improved. Special attention should be paid to Figure (6). While it is assumed that the information plotted there is essential, the layout should be improved to make it more legible, so to speak. Alternately, a good idea would be to condense some figures in one if the Authors consider it feasible.
- And finally but by far no less important. This reviewer assumes that the study is quite interesting yet difficult to be generalized to other locations. However, it would improve a bit more the quality of the document, otherwise a quite good analyzed report on typhoon surge and wave interaction, if the authors might provide with a final conclusion on how the results/methodology/both might or might not be applied to other general conditions. For example: the final information of the 0.15m wave set up is such a valuable one. Do the Authors could provide with some information about how that reference value should be corrected in terms of typhoon intensity/latitude/etc?
Round 2
Reviewer 1 Report
The manuscript has been significantly improved and I propose it forpublication in Applied Sciences.
The authors should check the equation 16 - is missing!
Author Response
We fix the bug.The equation 16 is displayed correctly now.